# Simultaneous Quantification of Organic Acids in Tamarillo (*Solanum betaceum*) and Untargeted Chemotyping Using Methyl Chloroformate Derivatisation and GC-MS

**DOI:** 10.3390/molecules27041314

**Published:** 2022-02-15

**Authors:** Chris Pook, Tung Thanh Diep, Michelle Ji Yeon Yoo

**Affiliations:** 1The Liggins Institute, The University of Auckland, Auckland 1142, New Zealand; 2School of Science, Auckland University of Technology, Auckland 1010, New Zealand; tung.diep@aut.ac.nz (T.T.D.); michelle.yoo@aut.ac.nz (M.J.Y.Y.)

**Keywords:** fruit, ripening, citric acid, itaconic acid, metabolomics, metabolite profiling

## Abstract

Sixteen organic acids were quantified in peel and pulp of Amber, Laird’s Large and Mulligan cultivars of tamarillo using GC-MS. Fourteen of these compounds had not previously been quantified in tamarillo. An untargeted metabolomics approach was used in parallel to identify and quantify 64 more metabolites relative to the internal standard, indicating abundances of glutamic acid, pro-line, aspartic acid and γ-aminobutyric acid as well as lower concentrations of several other essential fatty acids and amino acids. The main findings were that total organic acid concentration was significantly higher (*p* < 0.05) in pulp than in peel, with the highest concentration seen in Mulligan pulp (219.7 mg/g DW). Remarkably, after citric acid, the potent bactericide itaconic acid was the second most abundant organic acid. At least 95% of organic acids in tamarillo were one of these two acids, as well as cis-aconitic, malic and 4-toluic acids. Differences between cultivar chemotypes were as substantial as differences between tissues. These results suggest that the bitter flavour of the peel does not result from organic acids. The combination of targeted and untargeted metabolomics techniques for simultaneous qualitative and quantitative investigation of nutrients and flavours is efficient and informative.

## 1. Introduction

Tamarillo or tree tomato, *Solanum betaceum* (Cav.), is a small tree or shrub belonging to the Solanaceae family. The plant is native to the Andes but has been transplanted around the world. The various cultivars of tamarillo yield fruit that are oval, 4–10cm in length, and ripe colour from golden-yellow to orange, red and purple. The fruit contain high levels of nutrients including carotenoids, anthocyanins, phenolic compounds, vitamins and flavonols [1,2,3,4]. Tamarillo are also a rich source of dietary minerals while being low in caloric content [5]. Besides their nutritious properties, consumption of tamarillo has been shown to produce antioxidant, anti-inflammatory, anti-obesogenic and chemopreventive effects in rodent and human cancer cell models [6,7].

The flesh of tamarillo can be consumed raw. As with the common tomato, *Solanum lycopersicum*, the peel, skin or epicarp of tamarillo is edible but is usually discarded due to its bitter taste and waxy texture. When cooked, however, the skin can be incorporated into dishes and consumed and may even have unique nutritious properties [8]. In Aotearoa New Zealand, tamarillo fruit are grown domestically and commercially. Only Aotearoa New Zealand, Colombia and Australia export tamarillo. Tamarillo fruit are considered to be an under-studied and under-developed resource and are the subject of growing interest in food and non-food product formulations [9].

Gas Chromatography with Mass Spectrometry (GC-MS) can be used to provide high-throughput profiling of metabolomes in an untargeted fashion to reveal patterns of metabolite expression. Villas-Bôas et al. [10] proposed that such metabolites can be analysed in plants by GC-MS, with one single analysis by converting amino and non-amino OAs to volatile derivatives using derivatization with either alkyl chloroformates (ACF) or with silylation reagents [11]. Many compounds important to nutritional or flavour profiles of food may be derivatized by ACF and have been quantified using this technique, including fatty acids [12], phenolic acids [13], amino acids and non-amino acids [10], phytohormones [14] and diverse metabolites from the gut microbiome [15]. This information is of great interest to the study of human nutrition, to food scientists and to agricultural scientists, as indicators of plant health, development crop yield per area, disease resistance and shelf-life [16]. For example, Nascimento et al. [17] used MCF derivatization to detect 76 metabolites from seven different classes in bananas harvested from locations either proximate to or distant from native forest stands and revealed that the proximate samples exhibited increased concentrations of nutrients, such as γ-aminobutyric acid (GABA) and ω-3 polyunsaturated fatty acids.

The small molecules amenable to ACF-based analysis, such as organic acids (OA), are generally the same compounds responsible for taste preferences of consumers, determining flavour and aroma [18,19]. During ripening in many fleshy fruit, carbohydrates are broken down to sugars to fuel metabolism and growth, driving concomitant increases in other primary metabolites, such as organic acids [20]. As intermediates in the tricarboxylic acid cycle, and as precursors to many other functional molecules, organic acids are particularly important for fruit growth, development and quality. Such compounds are now of significance as markers of the authenticity and quality of food products [21,22]. For example, organic acids contribute to the sour flavour of fruit and can intensify or suppress the perceived sweetness [18,23]. However, the organic acid profile of fruit is also of interest due to their antibacterial activity and plant-derived organic acids have been proposed as a possible substitute for antibiotics in farming [24].

Although citric and malic acids have been quantified in five cultivars of tamarillo sourced from Spain and Ecuador, respectively, the rest of the organic acid profile of tamarillo has not been studied in detail. Our own group has recently reported the content of sugars, phenolic compounds, anthocyanin and carotenoid pigments, ascorbic acid and α-tocopherol and amino acids using targeted assays [3,4,5]. We have also carried out analysis of the volatile profile of freeze-dried tamarillo [25] and some characterization of the volatile profile and presence of phenolic compounds has been reported elsewhere in the literature [26,27,28,29,30]. In the former research we discovered that extracts of both tamarillo peel and pulp exhibited strong antibacterial activity and propose that both may prove useful as a natural preservative when added to food products. The use of peel for this purpose may also represent a valuable use of a material commonly treated as waste. To better comprehend the role that organic acids play in tamarillo, and how they contribute to antibacterial activity, we have applied methyl chloroformate (MCF) derivatization and GC-MS to compare metabolite profiles of organic, amino and fatty acids in the peel (epicarp) and pulp (mesocarp) of three different tamarillo cultivars (Amber, Laird’s Large and Mulligan) grown in Aotearoa New Zealand. Untargeted metabolomic analysis provides no information about the absolute concentration of each metabolite, which is critical to understand the functionality of biologically active compounds. Here we have deployed a hybrid quantitative approach using both internal and external standards to absolutely quantify the sixteen most abundant OAs in tamarillo whilst simultaneously capturing the metabolic profile amenable to ACF derivatization.

We deploy a hybrid approach using both internal and external standards which allows us to explore the metabolome amenable to ACF derivatization in an untargeted fashion while simultaneously providing absolute quantification of the sixteen most abundant OAs down to the parts per billion range.

## 2. Results

### 2.1. Targeted Analysis Validation

Most of the sixteen organic acids spiked into tamarillo samples showed satisfactory recoveries from 70 to 120% (Appendix A). Example chromatograms for a sample of Mulligan pulp and a 50 mg/L organic acid standard can be seen in Figure 1. Triplicate spike and recovery measurements showed good reproducibility with mean Relative Standard Deviations across the sixteen targets of 5.27% and standard deviation of 3.85%. The exceptions were the most abundant OAs in the tamarillo samples, citric and itaconic acids, with mean recoveries of 169% and 175%, respectively. These values were consistent, however, with RSD of 1.0% and 1.6%. Lactic acid showed low mean recovery of 52%. All of the reported OA concentrations have been adjusted by the recoveries measured.

### 2.2. Targeted Analysis Results

Absolute quantitative analysis of sixteen OAs extracted from tamarillo by MCF derivatization and GC-MS showed that these compounds comprised an average (±SD) of 81.5 (±25.0) mg per gram dry weight (mg/gDW) of tamarillo peel. Mean ±SD OA content of pulp was 98.6 (±14.2) mg/gDW. Variation in total OA concentration between the cultivars was greater in peel (Relative Standard Deviation (RSD) 34%) than in pulp (RSD 14%). The total organic acid content differed significantly between some cultivars and tissues (Kruskal–Wallis Statistic = 18.950, *p* < 0.01). Mulligan pulp had the highest Total Organic Acid Content (TOAC) with 110.8 (±13.5) mg/gDW, followed by Amber pulp with 8% less than this value. Mean TOAC for Laird’s Large was significantly lower, 25% less than that of Mulligan and 19% less than that of Amber. Mulligan peel TOAC was 59% less than its pulp and Amber peel was 43% less than pulp. There was no significant difference be-tween the TOAC of Laird’s Large pulp and peel and its peel had the highest value for this tissue type, 88.8 ± 31.5 mg/gDW. Laird’s Large was the only cultivar where peel TOAC exceeded that in pulp.

The mean concentration of each OA quantified in tamarillo decreased in the order: citric > itaconic > malic > cis-aconitic > 4-toluic > 2-hydroxybutyric > malonic > succinic > fumaric > 3-hydroxybenzoic > lactic > vanillic > syringic > adipic > trans-cinnamic > ben-zoic (Figure 2). Citric, itaconic, cis-aconitic, malic and 4-toluic acid together comprised >97% of the total organic acid quantified in each sample, with mean concentrations of 39.3 (±10.7), 19.6 (±7.5), 10.0 (±4.1), 8.7 (±3.5) and 1.4 (±0.7) mg/gDW, respectively. Concentrations of syringic, vanillic, adipic, trans-cinnamic and benzoic were <0.01 mg/gDW. Neither adipic, syringic nor trans-cinnamic acids were detected in Laird’s Large pulp and benzoic acid was not detected in Amber peel.

There were highly significant differences in the concentration of all OAs between cultivars and between tissues. Concentrations of most OAs were significantly higher in the pulp of tamarillo than in their peel. Only fumaric, malonic, syringic and adipic acids were significantly higher in concentration in the peel of some cultivars than in pulp. The exception to this was Laird’s Large tamarillo, whose pulp and peel showed no differences in concentration of the five most abundant OAs.

### 2.3. Untargeted Analysis Results

Besides the 16 OAs targeted there were 64 features in the GC-MS data to which identities could be assigned using the NIST and MCF libraries including dinucleotides, hydroxy acids and their derivatives, cinnamic and hydroxycinnamic acids, benzoic acids and their derivatives, keto acids, sugar acids and their derivatives, carboxylic acid and their derivatives, fatty acids and fatty acid esters and amino acids and their derivatives. Of these, 55 were identified using the MCF library to level one of the Minimum Reporting Standards for Chemical Analysis (MRSCA) and the remaining nine were identified using the NIST library to level two. Of the latter, only 5-oxotetrahydrofuran-2-carboxylic acid proved to be significant in the analysis.

The compounds shown in Figure 3 were quantified relative to the recovery of the d_4_-alanine internal standard (Figure 3B). The twelve most abundant compounds, by mean relative peak area across all samples, were citric acid, glutamic acid, proline, aspartic acid, malic acid, 4-aminobutyric acid, asparagine, alanine, pyroglutamic acid, cis-aconitic acid and glutathione. Curiously, of the four most abundant compounds quantified by targeted analysis, only three appear in this list. The MCF derivative of itaconic acid was the 26th most relatively abundant feature.

The untargeted data were combined with the targeted data, standardized by subtracting the mean and dividing by the SD, and analysed by Principal Component Analysis (Figure 3A). The samples were almost perfectly resolved by tissue type on the first Principal Component (PC1), which explained 41.9% of the variance in the data, and by cultivar on PC2, which explained another 21.3% of the variance. No further PCs were considered. The only samples which weren’t resolved by PCA were the Amber and Mulligan peel samples, which clustered closely together in the lower left quadrant, and whose 95% confidence intervals overlapped. Non-standardized data for the twelve compounds detected in untargeted analysis that drive the differences on each PC (PCA loadings) are plotted by tissue and by cultivar in Figure 4 and Figure 5.

Of the twelve compounds responsible for differences between the metabolite profiles of pulp from the three cultivars and their peel, eleven were significantly different at *p* < 0.001 (Appendix A), whereas for citric acid *p* < 0.01. Concentrations of 4-aminobutyric acid (GABA), cis-aconitic acid, lysine, phenylalanine, glycine, isocitric acid and β-citryl-L-glutamic acid were typically elevated in pulp. The exceptions to this were citric acid, aconitic acid and 2-oxoglutaric acid concentrations in Laird’s Large pulp and peel samples, which were not significantly different. Only sinapic acid and anthranilic acid were depleted in pulp, relative to peel samples. Isocitric acid was not detected in Mulligan peel.

The twelve compounds most responsible for differences in metabolite profile between the three cultivars, as revealed on PC2, include asparagine, tyrosine, maleic acid, malonic acid, NADP/NADPH, benzoic acid, creatinine, coumaric acid, 5-oxotetrahydrofuran-2-carboxylic acid, octanoic acid, 2-methyloctadecanoic acid and cinnamic acid. All these exhibited significant differences between the tissues of different cultivars at *p* < 0.001, with the exception of NADP, which was significant at *p* < 0.05, and 5-oxotetrahydrofuran-2-carboxylic acid, which was not significant after post-hoc testing. Asparagine was elevated in the tissues of Amber and Mulligan tamarillo but depleted in that of Laird’s Large. Tyrosine concentrations in Amber samples were greatly elevated above those of other cultivars, for both peel and pulp. Maleic acid in Amber pulp and Mulligan pulp was substantially elevated above all other cultivar tissues, except Amber peel, which was intermediate. Malonic acid was significantly elevated in Amber and Mulligan peel. In contrast, malonic and maleic acids were significantly reduced in Laird’s Large pulp.

Neither 5-oxotetrahydrofuran-2-carboxylic acid nor 2-methyloctadecanoic acid were detected in any of the Amber samples. In addition, 2-methyloctadecanoic acid was not detected in Mulligan peel and cinnamic acid was not detected in either Mulligan tissue.

Figure 3C illustrates the relative contribution of each class to the total number of compounds identified. The amino acids and their derivatives were the most dominant chemical class which comprised 31% (30 compounds) in total. The fatty acids and their esters as well as carboxylic acids and their derivatives were the second most abundant groups, of 19% and 18%, respectively. Keto acids, sugar acids and their derivatives comprised 9% of the compounds observed in tamarillo.

## 3. Discussion

We applied methyl chloroformate derivatization and GC-MS to target sixteen organic acids for quantification in peel and pulp of three Aotearoa New Zealand tamarillo cultivars. The results showed that the potent bacteriocide, itaconic acid, was the second most abundant organic acid in tamarillo peel and pulp, after citric acid. These two acids, together with malic, cis-aconitic and 4-toluic acids, comprise more than 98% of the total organic acid content of tamarillo. We simultaneously acquired data on the tamarillo metabolome amenable to alkylation, yielding semiquantitative data for another 64 compounds. Our result show that the metabolomes of Amber and Mulligan peel are similar, while that of their pulp is somewhat differentiated. The metabolome of Laird’s Large tamarillo peel and pulp are more distinct from that of other cultivars and less so from each other.

Itaconic acid is a known bacteriocide from its role in the animalian immune response, where it is synthesised by alveolar macrophages as a defence against bacterial lung infections by inhibiting isocitrate lyase [31,32]. It has also been shown to play a role in mollusc microbial defences [33]. It is rarely mentioned in the literature as a component of fruit metabolism, although it has been previously reported in tomato and capsicum [34]. That we have measured it as being one of the most abundant OAs in tamarillo is remarkable because in fruit tissue citric and malic acids usually dominate [35]. Besides itaconic acid, the organic acid profile of tamarillo was dominated by citric, malic, cis-aconitic and 4-toluic acids. Citric acid naturally exists in a variety of fruits and contributes to their pleasantly sour taste. Malic acid is also known to be antimicrobial and growth of *Listeria monocytogenes*, *Salmonella gaminara*, and *Escherichia coli* O157 has been reported in some fruit juices [36]. The concentrations of citric and malic acids we have measured in tamarillo pulp, ranging from 68–81 mg/g DW citric and 6.4–13.8 mg/g DW malic acid, are similar to those reported elsewhere: Vasco et al. [29] analysed tamarillos similar to the Amber and Laird’s Large tamarillos from this study, sourced from two different countries. They found no difference between cultivars but reported citric acid concentrations in peeled fruit from Spain and Ecuador that are 191% and 351%, respectively, of our mean value, and malic concentrations that were 44% and 410% of our mean. Boyes & Strübi [28] measured mean citric acid concentrations 197% and malic acid concentrations 129% of our means, assuming a water content of 90%. Acosta-Quezada et al. [30] reported that mean citric acid concentrations in red and purple tamarillos were 72% and 86%, respectively, and malic acid concentrations were 65% and 67% of our values. Vasco et al. (2015) used HPLC for quantitative analysis, which is a less specific technique than GC-MS, so we suggest that their comparatively high concentrations are the result of methodological artefacts, possibly resulting from coelution of itaconic acid and citric acid.

Succinic acid showed higher concentration in the pulp than in the peel of all Aotearoa New Zealand tamarillo cultivars (Figure 1). This acid contributes to sour and astringent flavour and is often used as a flavouring agent. According to Mokbel and Hashinaga [37] an abundance of succinic acid could extend the shelf life of fruit (banana) due to inhibition of fungal growth. This study also indicated that malic and succinic acids could prevent the growth of food poisoning bacteria in vivo including Gram-positive species (*Bacillus subtilis*, *B. cereus*, *Staphylococcus aureus*) and Gram-negative species (*Salmonella enteritidis*, *E. coli*).

Water, methanol, ethanol and hexane extracts of tamarillo peel and pulp have shown antibacterial activity against Gram-negative (*E. coli, Pseudomonas aeruginosa*) and Gram-positive bacteria (*S. aureus* and *Streptococcus pyogenes*). In particular, water extracts of peel and pulp showed considerable activity against E. coli and methanolic extracts of peel and pulp against the Gram-negative species. The most abundant organic acids we have found in tamarillo are all highly polar and so soluble in water and methanolic extracts and would have been efficiently extracted from the tissues by these solvents. It is likely that they are at least partly responsible for the observed antibacterial activity.

It is notable that both Amber and Mulligan cultivars of tamarillo had lower concentrations of organic acids in their peel than in their pulp. The peel of tamarillo is rarely eaten fresh as it has a bitter taste. Our results indicate that this is not a result of the accumulation of organic acids but from another metabolite, such as the saponin glycoside solanine that gives the closely related bitter tomato (*Solanum aethiopicum*) its characteristic bitterness. Such compounds will not be amenable to methyl chloroformate derivatisation and so will be visible to this technique.

Using targeted LC-MS/MS analysis, our group has previously reported concentrations of several hydroxycinnamic acids (chlorogenic, caffeic, coumaric, ferulic) and hydroxybenzoic acids (gallic, ellagic) in the same tamarillo samples investigated here [3]. With the exception of coumaric acid we found no trace of these compounds in this analysis, likely due to the lower sensitivity of this GC-MS method. Concentrations of some phenolic acids have been reported in Malaysian tamarillo [38]. This study reported mean vanillic acid concentrations in tamarillo that were approximately five times that of their study. We observed p-coumaric acid in the untargeted analysis but only report relative abundance as the concentration was so low. Concurrent with our findings Mutalib et al. (2016) report this compound to be present at sub-ppm concentrations. Those researchers also reported concentrations of gallic, trans-ferulic and caffeic acids. However, we found no trace of these in our analysis.

Through untargeted analysis we have demonstrated that tamarillo are a source of many other metabolites that contribute to flavour and nutrition. These include essential fatty acids, linoleic acid and α-linolenic acid, the essential amino acids, valine, leucine, isoleucine, threonine and phenylalanine, and conditionally essential amino acids, cysteine, glutamine, glycine, proline and tyrosine. Our results compare well with other metabolite profiling studies that use derivatisation and GC-MS to reveal differences in the nutritional and flavour properties of foods. Nascimento et al. [17] used MCF derivatisation to profile the metabolites of bananas harvested from locations either proximate to or distant from native forest stands, as mentioned above. Hijaz et al. [14] used MCF derivatisation and GC-MS to study the altered expression of phytohormones in sweet oranges following exogenous application of GABA. They found changes in the relative concentrations of cinnamic acid, benzoic and coumaric acid which were observed to differ between tamarillo cultivars in this study. Coumaric and cinnamic acids are of relevance as the difference in colouration between Amber tamarillo and the two other cultivars studied here is a result of substantial reductions in concentration of red and purple anthocyanin pigments in Amber tissues [3]. Anthocyanins and other flavonoids are synthesised by the phenylpropanoid pathway from the amino acids phenylalanine and tyrosine. These are metabolised to cinnamic acid and p-coumaric acid, which is then metabolised further to yield a great diversity of natural products, including the flavonoid precursors to anthocyanin pigments. We found that tyrosine concentrations of Amber tamarillo tissues were approximately double those of the two other cultivars and the coumaric acid content of Amber pulp was more than double that of other pulp samples. Tyrosine is synthesized downstream of the shikimate pathway, which also provides the precursors to the other aromatic amino acids, phenylalanine and tryptophan [39]. This confirms the results of our previous analysis where we observed higher phenylalanine and tryptophan concentrations in Amber tamarillo than in the other cultivars [5]. The concentration of *p*-coumaric acid in Amber were also almost double those in other cultivars [3]. It is not possible to tell from single point analyses whether fluxes of metabolites have changed but we suggest that this accumulation of precursors in Amber tamarillo likely results from a genetic change in the anthocyanin synthesis pathway that results in a metabolic block. As a result the precursors to anthocyanin synthesis accumulate in the tissues of Amber tamarillo.

The absolute concentration of amino acids have recently been quantified in tamarillo by our group [5] and our results here largely agree with those. Eleven of the 22 most relatively abundant compounds measured here were amino acids or peptides with glutamic acid (umami, savoury) and aspartic acid (sour, umami) being second and fourth, respectively. Amino acids play a major role in determining flavour as well as possessing unique nutritional and bioactive properties. The others were relatively abundant amino acids and their flavours include proline (sweet), aspartic acid (sour, umami), GABA (sour, bitter), asparagine (tasteless), alanine (sweet), pyroglutamic acid (sour), glutathione (no flavour but enhances umami flavours), lysine (savoury) and valine (bitter), tyrosine (bitter) and leucine (flat to bitter) [40]. Of note amongst these is the inhibitory neurotransmitter, GABA, with demonstrated health promoting effects, including anti-hypertensive and immunity-enhancing properties [5,41].

The use of global metabolite profiling techniques involving derivatisation and GC-MS now appear to be in widespread use in the field of food science. There is a trend towards silylation derivatisation, as described above, as opposed to alkylation. Our results show that the technique has great potential to advance the analysis of organoleptic and bioactive properties of foods by revealing differences in the abundance of odour, flavour and nutritional compounds which are otherwise quantified through multiple, discrete analyses, with the increased costs and labour associated with that approach. For example, Song et al. [42] quantified volatile flavour compounds, amino acids, fatty acids and sugars by discrete analyses. We have quantified compounds from three of those four types of compounds using a single analysis. The discrete approach may suffer from poor comparative results between different analytical techniques, such as the ten-fold difference in citric acid concentration between traditional techniques and a silylation and GC-MS technique reported by Roessner et al. [43]. The method reported here shows a dynamic range that is much wider than many reported in the literature for similar analysis. This is a particularly important attribute of the method because the highest concentrations of OAs reported here (~50 mg/gDW) are one million times the lowest (~50 ng/gDW). This wide dynamic range is particularly important to analysis of nutrition and metabolism, where the concentration of those compounds can vary across an even wider range and coverage is largely determined by that parameter. There is no obvious reason for the low recovery of lactic acid in this analysis. In crude tissue homogenates, lactic acid may be biologically labile due to the role it plays in aerobic metabolism. However, as that lability is due to enzymatic metabolism, it is unlikely to be the cause of the effect measured here as those enzymes will have been denatured by the extraction solvent.

The two quantitative approaches implemented in this study are complimentary: While absolute quantification is more labour intensive to carry out than relative quantification, when studying novel or previously unstudied food products, such as the organic acid content of tamarillo reported here, it is necessary to establish the absolute values for establishing nutritive properties. Relative quantification is simpler to apply and is of particular use when carrying out comparative analysis of established food products where it is not necessary to establish the absolute quantity of flavour and nutritive compounds but their relative difference as a result of some experimental process or manipulation. For example, here we report the absolute values of sixteen OAs and reveal which ones contribute to the bioactive, nutritional, and organoleptic properties of the different tamarillo cultivars and tissues.

The MCF derivatisation has some limitations to its application to the analysis of organoleptic properties as many plants synthesise alkyl esters and the derivatisation converts carboxylic acids to alkyl esters. Therefore, it is not possible to differentiate between the carboxylic and esterified forms of these compounds. For example, the abundance of ethyl esters of fatty acids is a critical parameter in determining the quality of olive oil [44,45]. And so, the use of ethyl chloroformate to profile olive oil products would fail to distinguish between these endpoints and endogenous fatty acids. The same applies to a variety of compounds found in the metabolite profile of passionfruit, including ethyl cinnamate, propyl myristate, methyl benzoate, methyl dodecanoate, methyl hexadecanoate and methyl dihydrojasmonate [46]. In this study we have assumed that, where the library indicates that a compound is a methyl ester, the esterification was a result of derivatisation. As these esters are typically amenable to GC-MS without derivatisation, their original forms can be confirmed using conventional volatile sample introduction techniques, such as Thermal Desorption-GC-MS [25]. An alternative strategy used to circumvent this issue is to analyse samples in parallel with two different alkyl chloroformates [15].

To conclude, analysis of the metabolite profile of tamarillo, *Solanum betaceum*, by MCF derivatisation and GC-MS reveals an organic acid profile dominated by citric acid and, unusually, the potent bacteriocide itaconic acid. Lesser components include cis-aconitic, malic and 4-toluic acids, together with traces of eleven others. Significantly higher amount of total organic acid was found in pulp than in peel of all cultivars examined, with the highest seen in Mulligan pulp. Tamarillo exhibits antibacterial properties and may be a useful ingredient for increasing the shelf-life of commercial food products. The use of MCF derivatisation and GC-MS allowed absolute or relative quantification of 16 OAs, and 64 metabolites including amino acids and fatty acids. This shows the value of applying modern, analytical techniques that can simultaneously provide accurate, targeted quantification as well as untargeted information to characterise the chemotype, or chemical space of the subject. Targeted approaches fail to detect novel or unusual metabolites and so will fail to reveal the value of the product under investigation.

## 4. Materials and Methods

### 4.1. Chemicals

Methyl chloroformate and a linear alkane series from heptane to triacontane were obtained from Sigma Aldrich (Sigma-Aldrich Pty Ltd., Sydney, Australia). Optima LC-MS grade methanol and analytical grade pyridine, chloroform, sodium acetate and sodium hydroxide were from Thermo Fisher (Thermo Fisher Scientific Inc., Auckland, New Zealand). Ultrapure water was obtained from a Purite Select Fusion system (Total Lab Systems, Auckland, New Zealand). Analytical grade standards of 2-hydroxybutyric, 3-hydroxybenzoic, adipic, benzoic, *cis*-aconitic, citric, fumaric, itaconic, lactic, malic, malonic, 4-toluic, succinic, syringic, cinnamic and vanillic acids, as well as d_4_-alanine, were obtained from Sigma Aldrich (Sigma-Aldrich Pty Ltd., Sydney, Australia).

### 4.2. Sample Preparation and Extraction

Samples of Amber, Mulligan and Laird’s Large cultivars of tamarillo (*Solanum betaceum*) fruit at commercial maturity stage (21–24 weeks) were collected from growers in the Northland region of Aotearoa New Zealand’s North Island. Peel and pulp tissues were dissected from four replicate fruit from each cultivar, snap frozen with liquid nitrogen and lyophilized, ground to powder and stored at −80 °C until extracted.

Lyophilized, ground tamarillo powder (2–2.5 mg, ±0.01 mg) was placed in a 1.5 mL polypropylene Eppendorf tube to which was added 200 µL of MCF extraction and derivatization solution (MCFIS), comprising 1 M sodium hydroxide, methanol and pyridine in a ratio of 5:4:1 (*v*/*v*), and 10 mg L^−1^ d_4_-alanine as internal standard. Samples were vortexed for 30 s and incubated for 10 min at 4 °C with vortexing every 5 min. The samples were then centrifuged at 1500× *g* RCF for 10 min at 4 °C and a 100 µL volume of supernatant was transferred to a polypropylene autosampler vial and capped for derivatization.

### 4.3. Analytical Standards

Stock solutions of the OA standards were prepared in MCFIS with a concentration of 20 g L^−1^. A mixed standard solution was prepared by combination of appropriate volumes of each stock and dilution with MCFIS to a concentration of 800 mg L^−1^. This top standard concentration was then serially diluted 9 times with MCFIS to give ten standard concentrations ranging from 1.5625 to 800 mg L^−1^. A 100 µL volume of each standard was placed in a polypropylene autosampler vial and capped for derivatization.

### 4.4. Derivatisation

Derivatization reagents comprised pure MCF, chloroform and 1.5 M sodium acetate buffer pH 5.3. Automated derivatization was carried out using a Gerstel Multipurpose Sampler (MPS) based on a scaled-down version of Smart et al. [47]. Reagents were added in the order: two 10 µL aliquots of MCF, 100 µL of sodium acetate buffer and 100 µL of chloroform. The reaction mixture was agitated continuously between additions and for 20 s following the addition of chloroform.

### 4.5. Gas Chromatography with Mass Spectrometry

The GC-MS instrument was an Agilent 6890B GC and 5977B MSD (Agilent Technologies, Santa Clara, CA, USA). The carrier gas was helium with a constant flow rate of 1.1 mL min^−1^. A 1 µL volume was injected into a split/splitless injector at 280 °C operating in pulsed splitless mode with a pressure of 26.1 psi for 2 min. The column was a Phenomenex ZB-1701 column (Phenomenex NZ, Auckland, New Zealand, 35 m × 250 μm × 0.15 μm with a 5 m guard). The oven was held at 40 °C for 5 min then rose to 300 °C at 10 °C min^−1^ and held for 2 min for a total run time of 33 min. The temperatures of 275, 230 and 150 °C were set for the transfer line, source and quadrupoles, respectively. The mass spectrometer scanned from 38 to 400 *m*/*z* with a rate of 4 scans s^−1^. Agilent MassHunter GC-MS software (Agilent Technologies, Melbourne, Australia) was used to collect and analyse the data.

### 4.6. Quality Control and Validation

Contamination and carryover were controlled using procedural blanks containing 100 µL of MCFIS which were prepared, derivatized and analysed between every 10 samples and standards. An alkane series was analysed alongside samples and standards to generate a Kovats Retention Index for untargeted analysis. Recovery of the 16 OA targets was assessed by spike and recovery. Triplicate aliquots of Mulligan pulp and Laird’s Large pulp sample were spiked with a 10 µL volume of the 200 mg L^−1^ standard and triplicate blanks were spiked with 10 µL of MCFIS solution. The samples were homogenized, extracted and analysed as above.

### 4.7. Targeted Data Processing

Concentrations of the sixteen organic acids were quantified using Agilent MassHunter Quantitative software (Agilent Technologies, Melbourne, Australia). Peak areas were calculated for the reference ion for each target in each sample, standard and blank, and normalized to recovery of the internal standard. Standard curves were constructed using linear relationships between relative peak area and concentration. Linearity was lost at the higher concentrations of lactic, benzoic and adipic acids and so data points were progressively removed from the top end of the calibration until linearity was achieved with a coefficient of correlation >0.99. Concentrations of these compounds measured in tamarillo were within the range of the standard points used. Summary data for each calibration are shown in Appendix A.

### 4.8. Untargeted Data Processing

Features in the raw data were extracted and identities were assigned using two distinct methods. First, Automated Mass spectral Deconvolution and Identification Software (AMDIS) (National Institute of Standards and Technology, Gaithersburg, MD, USA) was used to identify chromatographic features and deconvolute their mass spectra. A library of Kovats Retention Indices and spectra for MCF derivatives of common metabolites, produced by derivatization and analysis of authentic standards, was searched to assign identities to features in the data, as described by Smart et al. [47]. This corresponds with level 1 of the Minimum Reporting Standards for Chemical Analysis (MRSCA) proposed by the Metabolomics Standards Initiative [32]. Second, Agilent Unknowns software was used to identify chromatographic features and deconvolute their mass spectra. Identities were assigned to features by comparing their spectra and Kovats Retention Indices with the 2014 National Institute of Standards and Technology Mass Spectral Reference Library (National Institute of Standards and Technology, Gaithersburg, MD, USA). Features identified by these techniques were quantified relative to the recovery of the internal standard peak using Agilent MassHunter Quantitative software. These identifications correspond with level 2 of the MRSCA [48].

### 4.9. Statistical Analysis

All analyses were carried out in quadruplicate and the results are presented as mean (standard deviation). For comparison between the compounds quantified in different cultivars and tissues of tamarillos, one-way analysis of variance (ANOVA) was applied using the statsmodels’ library v0.10.0 for Python [49] if heteroscedasticity could be imposed by transformation. If data could not be transformed appropriately for parametric testing, then the Kruskal–Wallis test was used instead. Tukey’s (HSD) post-hoc pairwise comparison was applied to parametric data and Conover’s test was used for nonparametric data using the scikit-posthocs’ library for Python [50]. Principal Component Analysis (PCA) was carried out using the scikit-learn library v0.20 for Python [51]. The results of the untargeted analysis were visualized by heatmap and hierarchical clustering using the MetaboAnalyst web interface [52].

## Figures and Tables

**Figure 1 molecules-27-01314-f001:**
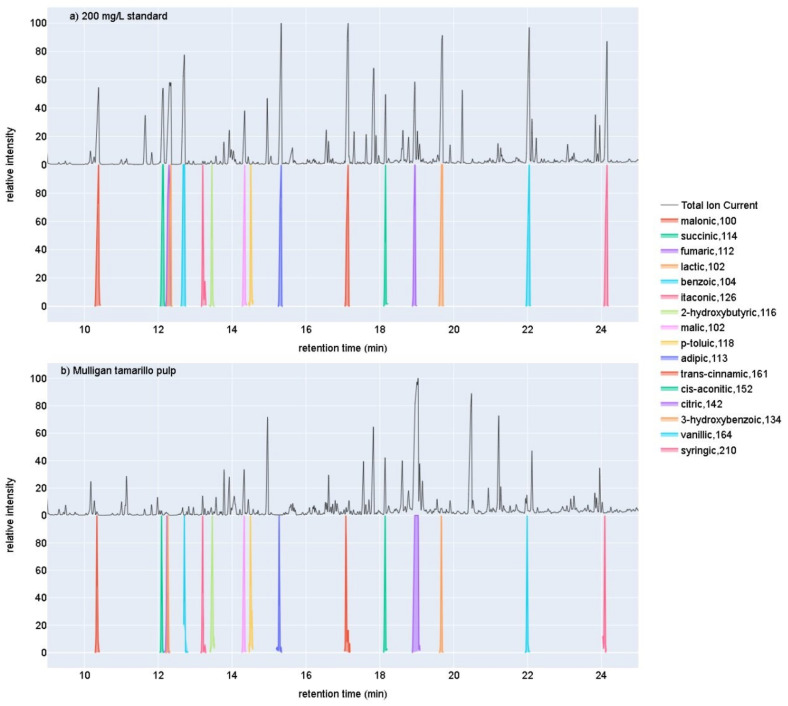
Example TIC chromatograms for (**a**) a 200 mg/L organic acid standard, and (**b**) a sample of Mulligan tamarillo pulp overlaid above the EIC chromatograms for each of the sixteen organic acids quantified here. The name of the organic acid for each EIC is followed by the *m*/*z* of the ion used for quantification.

**Figure 2 molecules-27-01314-f002:**
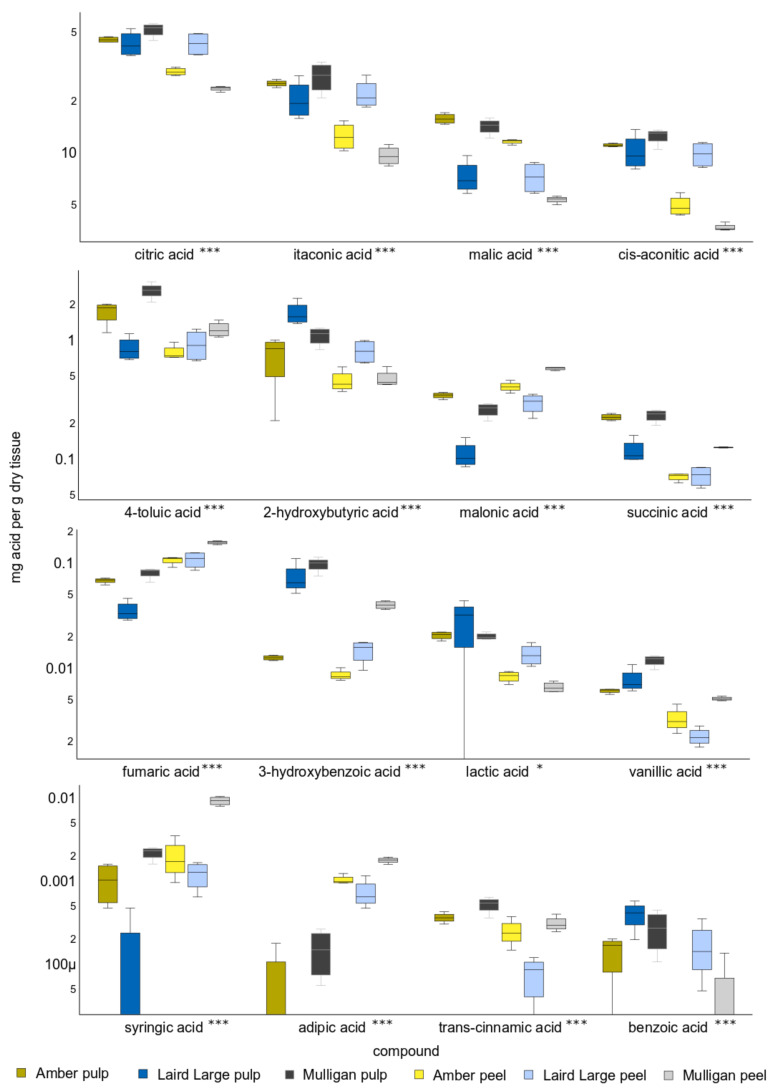
Boxplots showing the concentration of sixteen organic acids absolutely quantified in the peel and pulp of Amber, Laird’s Large and Mulligan tamarillo cultivars using MCF derivatization and GC-MS. The middle line of each box shows the median. Y-axis units are mg of organic acid per gram dry weight of tamarillo sample and are scaled logarithmically. Asterisks after the name indicate acids whose concentration differs significantly between cultivars and tissues. Single asterisks indicate *p* < 0.05 and triple asterisks *p* < 0.001.

**Figure 3 molecules-27-01314-f003:**
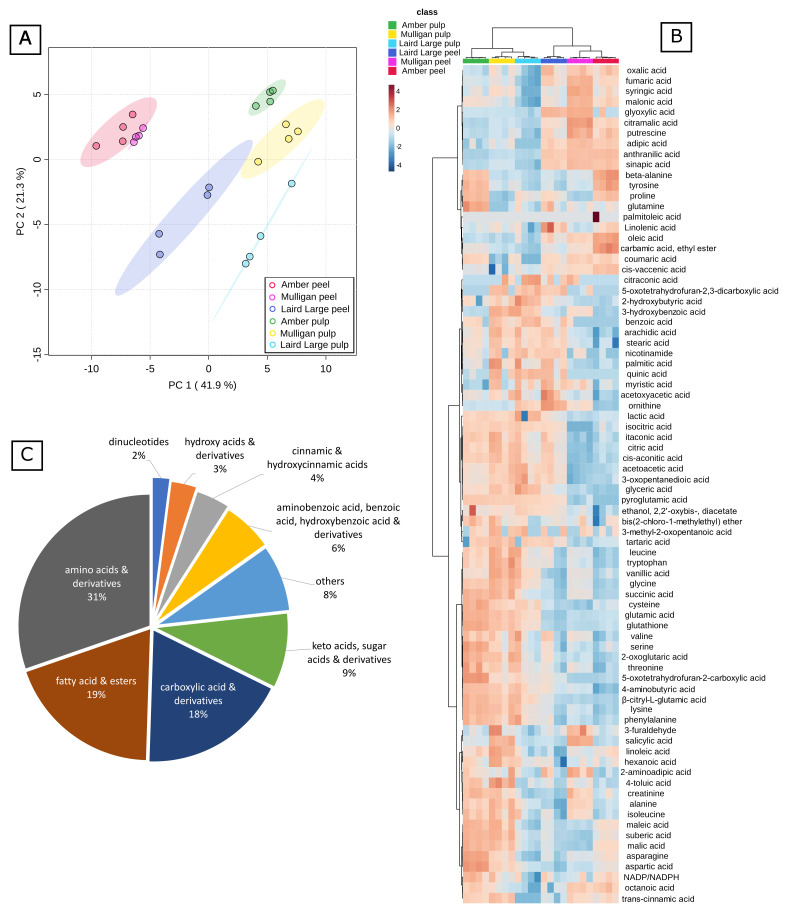
Statistical analysis and visualization of the combined data set. (**A**) shows Principal Component Analysis of the data; (**B**) is a heat map showing relative differences in each metabolite across the different sample types. Hierarchical Clustering has been applied to both axes. Note that the sample clustering resolves the different cultivars and tissue types perfectly; (**C**) is a pie chart showing the proportion of different metabolite class identities annotated here.

**Figure 4 molecules-27-01314-f004:**
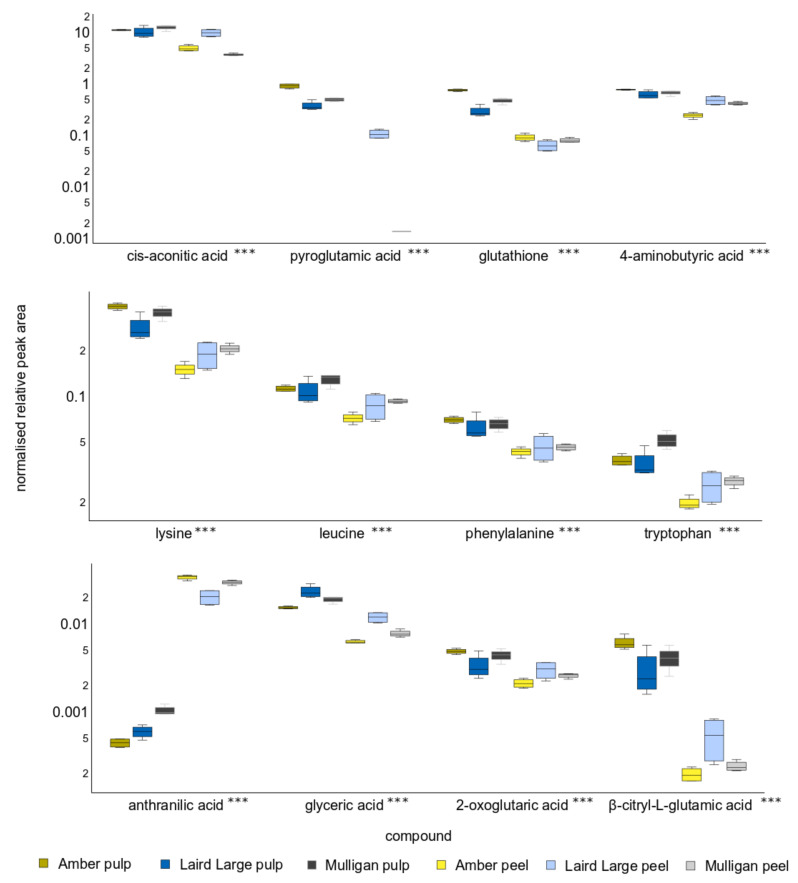
Relative peak area of the twelve compounds that contributed to the greatest differences on Principle Component 1, so discriminating between tamarillo pulp and peel. Y-axis units are peak area relative to recovery of the d_4_-alanine internal standard and are scaled logarithmically. The asterisks after the acid name indicates that the concentration of all compounds differed significantly between cultivars and tissues with *p* < 0.001.

**Figure 5 molecules-27-01314-f005:**
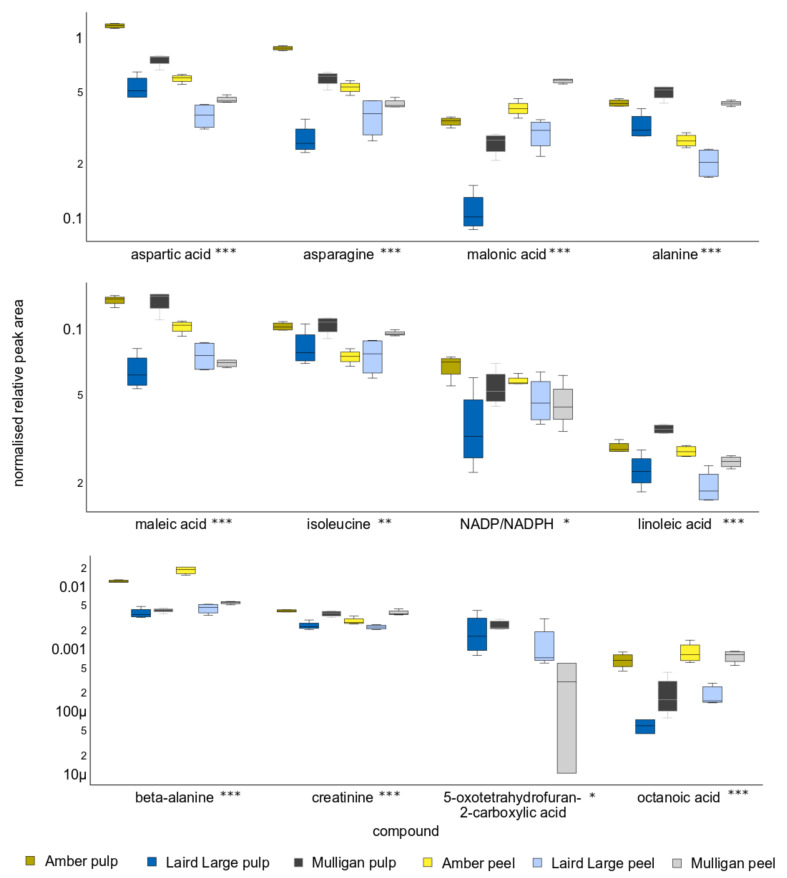
Relative peak area of the twelve compounds that contributed to the greatest differences on Principle Component Two, and so discriminating between the tissues of the Laird’s Large cultivar and the Amber and Mulligan ones. Y-axis units are peak area relative to recovery of the d_4_-alanine internal standard and are scaled logarithmically. Asterisks after the name indicate acids whose concentration differs significantly between cultivars and tissues. Single asterisks indicate *p* < 0.05, double indicates *p* < 0.01 and triple asterisks *p* < 0.001.

## Data Availability

Raw GC-MS data is available upon request.

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
