# Peer review of "Simultaneous Quantification of Organic Acids in Tamarillo (Solanum betaceum) and Untargeted Chemotyping Using Methyl Chloroformate Derivatisation and GC-MS"

_molecules, 2022, doi:10.3390/molecules27041314_

Round 1

Reviewer 1 Report

Dear Authors, your research paper is interesting. A lot of work you has been done in it. All your results are clear. In my opinion you should add more information what your results bring to people. Your work doesn't contain the conclusions what in my opinion is important.

Author Response

Thank you for your remarks. We have edited the introduction to emphasise the utility of the combined targeted and untargeted approach here. We have added emphasis on the antibacterial properties of tamarillo, as we report elsewhere. We have made the language stronger and simpler, and elaborated on the implications of our findings in the paragraph at the end of our discussion.

Reviewer 2 Report

This work reported simultaneous quantification of organic acids in tamarillo (So-lanum betaceum) and untargeted chemotyping using methyl chloroformate derivatisation and GC-MS. The topic is interesting. The results sounds successful. It may be accepted for publication.

Major comments:

Organic acids are one of the most important natural metabolites. Comprehensive analyses of organic acids and related metabolites is very important. Therefore, author may introduce some metabolomics analysis techniques to explore current methyl chloroformate derivated-GC/MS data, and add more comprehensive mathematic analyses results in the text.

Thus the topic may be focused on methyl chloroformate derivatisation GC/MS-based metabolomics analyses of organic acids in tamarillo.

Special comments:

Comprehensive GC-MS data including crude and processed data should be listed in the supporting files. It will help to understand the structures, contents, and other validations.

Author Response

This work reported simultaneous quantification of organic acids in tamarillo (Solanum betaceum) and untargeted chemotyping using methyl chloroformate derivatisation and GC-MS. The topic is interesting. The results sounds successful. It may be accepted for publication.

Thank you for your comments and your approval.

 Major comments:

 Organic acids are one of the most important natural metabolites. Comprehensive analyses of organic acids and related metabolites is very important. Therefore, author may introduce some metabolomics analysis techniques to explore current methyl chloroformate derivated-GC/MS data, and add more comprehensive mathematic analyses results in the text.

We are unclear as to what “comprehensive mathematic analyses results” are lacking from our manuscript. The mathematical calculations used to quantify the result presented here (recovery of internal standards and standard additions, linear regression for standard curves, means ±SD, percentage values) are pedestrian and ubiquitous in the metabolomics literature. Without specific examples we cannot make constructive revisions to the existing text.

Thus the topic may be focused on methyl chloroformate derivatisation GC/MS-based metabolomics analyses of organic acids in tamarillo.

We agree that the topic of the manuscript should focus on methyl chloroformate derivatisation GC-MS-based metabolomics to explore the organic acid content in tamarillo.

Special comments:

Comprehensive GC-MS data including crude and processed data should be listed in the supporting files. It will help to understand the structures, contents, and other validations.

We have added a summary table of processed data as Supplementary Information. Raw data files will be made available on request.

Reviewer 3 Report

Recommendation: Publish after minor revision.

This is an interesting, and well-designed manuscript. In this study, methyl chloroformate [MCF] derivatisation and GC-MS were applied to identify and quantify organic, amino and fatty acids in the peel and pulp of three different tamarillo cultivars. A lot of work has been done; however, a number of issues below should be addressed.

Please check the typo: for example, lines 31-32, 33, and114.

Please check grammar mistakes in the manuscript: Line 48-50, 63-65, 114-115.

Figure 1 and figure 4: significant difference should be included.

Representative GC-MS image is suggested to be shown in the study.

There was no mention of the limitations/weaknesses of the study.

Author Response

This is an interesting, and well-designed manuscript. In this study, methyl chloroformate [MCF] derivatisation and GC-MS were applied to identify and quantify organic, amino and fatty acids in the peel and pulp of three different tamarillo cultivars. A lot of work has been done; however, a number of issues below should be addressed.

Please check the typo: for example, lines 31-32, 33, and114.

We cannot find any typos on these lines. Line 114 is an empty row.

Please check grammar mistakes in the manuscript: Line 48-50, 63-65, 114-115.

 We cannot find any grammatical mistakes on these lines. Line 114 is an empty row.

Figure 1 and figure 4: significant difference should be included.

We have annotated each compound name with stars to indicate degree of significant difference. Attempts to label pairwise differences clutter the plot excessively and cannot be accommodated while maintaining readability. We have included a difference matrix as supplemental information to make this information available to readers.

Representative GC-MS image is suggested to be shown in the study.

Thank you for the suggestion, we have added annotated example chromatograms of a standard and a sample.

There was no mention of the limitations/weaknesses of the study.

We would like to draw the reviewer’s attention to lines 356-370, where we discuss the limitations of alkyl chloroformate derivatisation and GC-MS for analysing the organoleptic properties of tamarillo:

The MCF derivatisation has some limitations to its application to the analysis of organoleptic properties as many plants synthesise alkyl esters and the derivatisation con-verts carboxylic acids to alkyl esters. Therefore, it is not possible to differentiate between the carboxylic and esterified forms of these compounds. For example, the abundance of ethyl esters of fatty acids is a critical parameter in determining the quality of olive oil [48, 49]. And so, the use of ethyl chloroformate to profile olive oil products would fail to distinguish between these endpoints and endogenous fatty acids. The same applies to a variety of compounds found in the metabolite profile of passionfruit, including ethyl cinnamate, propyl myristate, methyl benzoate, methyl dodecanoate, methyl hexadecanoate and methyl dihydrojasmonate [50]. In this study we have assumed that, where the library indicates that a compound is a methyl ester, the esterification was a result of derivatisation. As these esters are typically amenable to GC-MS without derivatisation, their original forms can be confirmed using conventional volatile sample introduction techniques, such as Thermal Desorption-GC-MS. An alternative strategy used to circumvent this issue is to analyse samples in parallel with two different alkyl chloroformates [27]. 

We feel this is sufficient and appropriate.

Round 2

Reviewer 1 Report

Now, it looks much better. Thank you. Good luck!